# Visual Function in Children with GNAO1-Related Encephalopathy

**DOI:** 10.3390/genes14030544

**Published:** 2023-02-22

**Authors:** Maria Luigia Gambardella, Elisa Pede, Lorenzo Orazi, Simona Leone, Michela Quintiliani, Giulia Maria Amorelli, Maria Petrianni, Marta Galanti, Filippo Amore, Elisa Musto, Marco Perulli, Ilaria Contaldo, Chiara Veredice, Eugenio Maria Mercuri, Domenica Immacolata Battaglia, Daniela Ricci

**Affiliations:** 1Pediatric Neurology Unit, Fondazione Policlinico Universitario A. Gemelli IRCCS, 00168 Rome, Italy; 2National Centre of Services and Research for the Prevention of Blindness and Rehabilitation of Visually Impaired, IAPB Italia ONLUS, 00168 Rome, Italy; 3Ophthalmology Unit, Fondazione Policlinico Universitario A. Gemelli IRCCS, 00168 Rome, Italy

**Keywords:** neurovisual competences, GNAO1 mutation, visual function

## Abstract

Background: GNAO1-related encephalopathies include a broad spectrum of developmental disorders caused by de novo heterozygous mutations in the GNAO1 gene, encoding the G (o) subunit α of G-proteins. These conditions are characterized by epilepsy, movement disorders and developmental impairment, in combination or as isolated features. Objective: This study aimed at describing the profile of neurovisual competences in children with GNAO1 deficiency to better characterize the phenotype of the disease spectrum. Methods: Four male and three female patients with confirmed genetic diagnosis underwent neurological examination, visual function assessment, and neurovisual and ophthalmological evaluation. Present clinical history of epilepsy and movement disorders, and neuroimaging findings were also evaluated. Results: The assessment revealed two trends in visual development. Some aspects of visual function, such as discrimination and perception of distance, depth and volume, appeared to be impaired at all ages, with no sign of improvement. Other aspects, reliant on temporal lobe competences (ventral stream) and more related to object–face exploration, recognition and environmental control, appeared to be preserved and improved with age. Significance: Visual function is often impaired, with patterns of visual impairment affecting the ventral stream less.

## 1. Introduction

The GNAO1 gene encodes the α subunit of heterotrimeric guanine nucleotide-binding protein (Gi protein), a membrane protein that is involved in signal transduction and is widely expressed in the central nervous system [1]. Gi proteins modulate inhibitory signaling of many neurotransmitters, including GABA, adenosine and dopamine [2,3]. The Gi protein effects are the inhibition of adenylyl cyclase, the decrease in the production of cyclic adenosine monophosphate (cAMP), the inhibition of the N-type (Cav2.2) and P/Q-type calcium channels or the prevention of neurotransmitter release at the synaptic level [2].

The pathophysiologic mechanisms that underlie the aforementioned genetic movement disorders are still to be satisfactorily explicated.

In spite of that, the co-occurrence of manifold neurologic manifestations such as hyperkinesia and epilepsy could be explained by a series of putative disease mechanisms, which include altered neuronal excitability, presynaptic autoinhibitory effects and impaired modulation or transduction of transmembrane signaling.

Mutations in this gene have been reportedly associated with several clinical conditions, such as epileptic disorder, movement disorders (MDs) and developmental delay, isolated or in combination [4,5]. Different authors have recently reported that epileptic encephalopathy is, in most cases, related to GNAO1 loss-of-function (LOF) variants, while manifestations of MDs, with or without epilepsy, are predominantly related to gain-of-function (GOF) heterozygous variants [6].

The types of epilepsy so far reported range from developmental and epileptic encephalopathies (infantile epileptic spasm syndrome with hypsarrhythmia, Othahara syndrome and malignant migrating epilepsy) to focal or combined (focal and generalized) epilepsy with good response to antiseizure medicaments (ASMs) [3,7,8,9,10,11].

MDs generally include hyperkinetic syndromes, such as chorea, ballism, orofaciolingual dyskinesia and dystonia.

The onset of MDs is variable, but it usually takes place between the first months of life and the first decade. Symptoms are often preceded by pronounced hypotonia. The course of symptoms is fluctuating, and the response to medical treatment is frequently insufficient. Dyskinesia status can be unexpectedly triggered by fever, inflammation or pain, with violent mixed hyperkinesis in some cases. During exacerbation, patients may require intensive care admission for hyperthermia, rhabdomyolysis, dehydration and pre-renal failure [12]. Deep brain stimulation implantation in patients with recurrent hyperkinetic status and poor response to chronic medical treatment resulted in significant improvement in terms of symptom intensity and frequency of intensive care admission (12). Literature data show that both subthalamic and pallidal deep brain stimulation can be effective in GNAO1-associated dystonia and that implantation should be promptly considered in patients with sustained refractory GNAO1-associated dystonia [13].

Most patients mainly present phenotypes between epilepsy and movement disorders, whereas a small number of patients show both epilepsy and movement phenotypes equally.

Cognitive delay and poor verbal skills are also reported in both early- and late-onset encephalopathy, with severity varying from profound to mild [7]. Due to this severe clinical pattern, it is difficult to perform a structured cognitive assessment in all patients, and most case series only report a descriptive assessment of the cognitive status [7].

Recently, Graziola et al. [14] proposed, for the first time in this population, the assessment of cognitive and communication profile using eye-tracking technology (ETT), showing that motor and cognitive development may be highly discordant and that normal fluid intelligence may coexist with lack of any motor milestone.

Less research has been reported on visual performance, as structured assessments cannot be easily performed in children with major neurological and verbal impairment [15,16]. We developed an adapted a protocol for the assessment of visual function that proved to be reliable in describing visual competences in infants and children with different genetic conditions, with or without epilepsy.

The use of this protocol evidenced a strong correlation between early impairment of visual function and cognitive competences in epileptic encephalopathies such as West syndrome [17] and Dravet syndrome [18].

The aim of this study was to describe neurovisual competences in children with GNAO1 deficiency in order to provide additional data for the phenotypic characterization of the disease spectrum, with a specific focus on determining if visual skills are less impaired than motor competences.

## 2. Materials and Methods

### 2.1. Study Population

An observational study was performed on seven patients with GNAO1 deficiency disorders. We recruited seven children (four males and three females) from 2 to 8 years (mean age of 4.8 years), who were followed up at Gemelli Hospital Foundation, Bambino Gesù Hospital and Umberto I Hospital in Rome, and Meyer Hospital in Florence.

Five of the seven patients from this population (Pt 1, Pt 2, Pt 3, Pt 6 and Pt 7, respectively) had also been included in a previous study [14] (Pt 2, Pt 3, P 4, Pt 5 and Pt 6) to assess cognitive and communication profile using ETT.

#### Genetic Data

All patients had pathogenic GNAO1 variants (Table 1), confirmed with a custom enrichment panel for more than 100 genes related to pediatric MDs or a panel of next-generation sequencing for epilepsy.

### 2.2. Study Design

All patients underwent neurological and visual function assessment, ophthalmological evaluation and behavioral observation (quality of social skills and verbal ability).

#### 2.2.1. Neurological Assessment

The gross motor function classification system (GMFCS) was used to classify each child’s motor ability, with skill levels ranging from I to V, by means of observation [19]. This classification system categorizes five different levels of functioning. Distinctions among the levels are based on the child’s functional abilities related to gross motor movement.

Descriptions for each level: Level I—patient walks without limitations; Level II—patient walks with limitations; Level III—patient walks using a hand-held mobility device (canes or crutches); Level IV—self-mobility with limitations or self-powered mobility; Level V—patient transported in a manual wheelchair.

#### 2.2.2. Visual Function Assessment

The visual function assessment included the assessment of various aspects of visual function, such as ability to fix and follow, saccades, visual acuity, visual fields and attention at distance, contrast sensitivity and stereopsis, which are known to mature progressively from the first months of life. Ocular motility was also observed, and nystagmus, strabismus or abnormal ocular movements were recorded.

Ability to fix was assessed by observing the ability of the patient to fix on a high-contrast target (black/white or colored). Fixation is considered stable if it lasts for 3 s, unstable if it is shorter and absent if it cannot be elicited [17].

Tracking was assessed by observing the ability of the infant to follow a high-contrast target (black/white or colored) horizontally, vertically and in a full circle. Tracking is considered complete when covering the whole arc, incomplete when covering more than 50% of it, brief when covering less than 50% and absent when it cannot be elicited.

Attention at distance was tested by moving a colored toy (about 8–10 × 8–10 cm) backwards in a small arc away from the child. The maximum distance at which the child paid attention to the toy was recorded. At three months post-term age, the span covered ought to be 3 mt [20].

Saccadic movements were assessed using one target per hand. The child’s attention was alternatively drawn to the targets, horizontally (right and left) and vertically (up and down). The procedure was repeated twice on each side, noting if besides the eyes, the head was also involved in the movement.

Visual acuity was assessed binocularly by means of the Teller acuity card procedure [21,22]. This method is based on an inborn preference for a pattern (black and white gratings of decreasing stripe widths depicted on cards) over a uniform field. The location of the left/right position of the test stimulus randomly varies. An observer judges the patient’s reaction to the location of the test stimulus on the basis of eye and head movement. The threshold of acuity is taken as the minimum stripe width to which the subject consistently responds. Acuity values were expressed in minutes of arc (or cycles per degree) and were compared to age-specific normative data reported in the literature [21,22].

Binocular visual fields were assessed using kinetic perimetry, according to the technique described in detail by van Hof-van Duin [20]. The apparatus consists of two 4 cm wide black metal strips, mounted perpendicularly to each other and bent to form two arcs, each with a radius of 40 cm. The perimeter is placed in front of a black curtain concealing the observer, who can watch the patient’s eye and head movements through a peep hole. Subjects are held sitting or lying in the center of the arc perimeter, their chin supported. During central fixation on a 6° diameter white ball, an identical target is moved from the periphery towards the fixation point, along one of the arcs of the perimeter, at a velocity of about 3°/s. Eye and head movements towards the peripheral ball are used to estimate the outline of the visual fields. Age-specific normative data for full-term and pre-term infants are available [20].

Contrast sensitivity was assessed using the Hiding Heidi low-contrast test. It consists of four cards, one blank and the other three with a face on both sides, with contrast reducing from 100% to 25%, 10%, 5%, 2.5% and 1.25%. The test is administered moving the picture and the blank card at the same speed, usually horizontally. The side that the child looks at is noted as a response. The level of contrast sensitivity consists in the least contrasted picture that the child looks at.

Stereopsis was assessed using the Frisby stereotest, used to assess stereo vision at closer distances, requiring eye convergence [23,24]. The subject’s task is to detect a circle containing a pattern of geometric objects (the target) visible within a mosaic of similar geometric shapes. The target and the background are printed on the opposite sides of a Perspex plate and thus differ in their physical depth. The angular disparity depends on the thickness of the plate and on the distance from the observer.

#### 2.2.3. Ophthalmological Evaluation

The exam of the anterior segment was performed by a handheld slit lamp; fundus examination was performed by means of indirect binocular ophthalmoscopy with 28- and 20-diopter lenses. Cicloplegic refraction was performed by means of autorefraction. Second-level ophthalmological examination (OCT; retinography) was scheduled according to the level of patient’s cooperation.

#### 2.2.4. Behavioral Observation

Personal social skills (smiling, facial mimicry and communicative intentionality) and verbal ability (language reception and production) were evaluated in all children during clinical examination.

#### 2.2.5. Anamnestic Data Collection

Data on the characteristics of MD, epilepsy, neuroimaging findings and type of gene mutation were collected for all patients.

With regard to MD, the data collected included age at onset, phenomenology classification and treatment.

As for epilepsy, they included age at onset, semiology and frequency of seizures, recurrence of epilepticus, response to ASMs and epilepsy types, classified according to the ILAE criteria [25,26].

Verbal ability was evaluated considering the response to simple tasks, such as everyday object recognition with gaze orienting or pointing.

For all patients, correlations between neurovisual skills and other aspects of the clinical phenotype, of the neuroradiological features and of the type of genetic mutation were attempted.

## 3. Results

### 3.1. Neurological Data

All patients had hypotonia. Four had acquired head control (at 1.5 years of age). Of these four patients, one patient had acquired complete control (at 3 years of age) and the remaining three only partial trunk control. One patient walked with support, while the remaining six had not acquired walking. The gross motor function classification scale (GMFCS) levels were V for one patient and IV for the remaining six (Table 1).

The assessment of fine motor skills showed that all patients performed reaching movements. Two also performed prehension.

The assessment of personal and social skills showed that all patients were able to smile and performed good facial mimicry and communicative intentionality. All seven patients had good verbal comprehension, and only two patients were able to produce single words.

#### 3.1.1. Movement Disorders

All patients presented with hyperkinetic movement disorders, with three patients displaying chorea; three patients, dystonia; and one patient, ballism. The age at onset was within the 1st year in four patients; at 2 years, in two patients; and at 3 years, in the remaining one. Two patients presented with severe dyskinetic status and needed intensive care. The response to pharmacological therapy was good in five patients and poor in two patients. One of these two patients was a candidate for deep brain stimulation.

#### 3.1.2. Epilepsy

Four out of seven patients presented with epilepsy. One (Pt 4 in Table 1) presented with early-onset epileptic encephalopathy. The onset of seizures was at 1 month and was characterized by long migrating focal seizures, motor and non-motor, daily or weekly. Treated with polytherapy, they stopped at 14 months of age (more details in Table 1). Focal SE with poor symptoms appeared in the first year of life. At the moment of the study, Pt 4 was in monotherapy. The remaining three patients presented with focal epilepsy with onset after 20 months of age. All of them reported sporadic, focal motor and focal-to-bilateral tonic clonic seizures. Two patients were free from seizures at the time of evaluation (Pt 7 had been in monotherapy and seizure free since 6 years of age; Pt 6, without antiepileptic treatment, had been free from seizures since 3 years of age). The remaining patient had partial control in monotherapy.

**Table 1 genes-14-00544-t001:** Neurological data.

		Pt 1	Pt 2	Pt 3	Pt 4	Pt 5	Pt 6	Pt 7
Age/Sex		2 y/F	3 y/F	3 y/M	5 y/F	6 y/M	7 y/M	8 y/M
Mutation		p.Glu 246Lys	p.Glu 237Lys	p.Glu 246Lys	p.Gly 203Arg	p.Ser 47Gly	p.Gly 203Arg	p.Arg 209Cys
Milestones and motor competences	Head control	Yes	Yes	Yes	No	No	No	Yes
	Trunk control	Partial	Partial	Partial	No	No	No	Yes
	Deambulation	No	No	No	No	No	No	Yes, with support
	Reaching	Yes	Yes	Yes	Yes	Yes	Yes	Yes
	Prehension	No	No	No	No	Yes	No	Yes
GMFCS level		IV	IV	IV	IV	IV	V	IV
Hypotonia		Yes	Yes	Yes	Yes	Yes	Yes	Yes
Movement disorder	Type/age at onset	Chorea/8–9 m	Ballism/2 y	Chorea/8 m	Chorea/2 y	Chorea, dystonia/5 m	Dystonia/3 y	Dystonia/4–5 m
	Dyskinesia/dystonic status	No/no	No/no	No/no	Yes/no	No/yes	No/no	No/no
	Treatment	TBZ+ baclofen+ nitrazepam	Trihexyphenidyl+ TBZ	Trihexyphenidyl	Clonazepam+ TBZ	TBZ+ baclofen+ nitrazepam	TBZ	TBZ+ Clonazepam
Epilepsy	Presence/age at onset type of seizures/type of epilepsy	No	No	No	Yes/1 m focal seizures/early-onset encephalopathy	Yes/3 y focal motor/focal epilepsy	Yes/3 y focal motor/focal epilepsy	Yes/20 m focal motor/focal epilepsy
	SE	No	No	No	Yes	No	No	No
	ASM	No	No	No	FB, CBZ+ CLB, TPM+ CLB, TPM	TPM	No	CBZ
	Seizure control	**-**	**-**	**-**	Seizure free since 14 m with poly ASM	Not complete	Seizure free since 3 y	Seizure free since 20 m
MRI features		Not performed	Normal	Normal	Normal	Normal	Normal	Periventricular white matter hyperintensity
Sleep disorder		No	No	Yes	No	No	No	No
Personal social skills	Smiling	Yes	Yes	Yes	Yes	Yes	Yes	Yes
	Gaze engagement	Yes	Yes	Yes	Yes	Yes	Yes	Yes
	Facial mimicry	Yes	Yes	Yes	Yes	Yes	Yes	Yes
	Communicative intentionality	Yes	Yes	Yes	Yes	Yes	Yes	Yes
Verbal ability	Comprehension	Yes	Yes	Yes	Yes	Yes	Yes	Yes
	Production	No	No	Yes	No	Yes	No	Yes

Y, year; F, female; M, male; m, months; TBZ, tetrabenazine; SE, status epilepticus; ASM, antiseizure medication; FB, phenobarbital; CBZ, carbamazepine; CLB, clobazam; TPM, topiramate.

### 3.2. Visual Function Assessment

All patients completed the assessment. Details are shown in Table 2.

Ocular motility was abnormal in all patients presenting with ocular dyspraxia in six of the seven children, and strabismus and nystagmus in the remaining one.

Ability to fix was present in all patients, was stable in six of the seven patients and was unstable in the remaining one.

Tracking in a horizontal arc was complete in all patients. Tracking in a vertical arc was complete in three patients and incomplete in the remaining four patients. Tracking in a circle was complete in three patients, brief in three patients and absent in the remaining patient.

Attention at distance was normal in six patients and reduced in the remaining one.

Saccadic movements were present horizontally in six of the seven patients and were absent in the remaining one. Vertical saccades were present in three patients and absent in the remaining four patients.

Visual acuity was less than 3/10 in all patients. Six of the seven patients presented with visual acuity between 1 and 2.5/10, while the remaining patient presented with 1/50.

Binocular visual fields were normal in one patient, restricted in five patients and not evaluable in the remaining patient.

Contrast sensitivity was reduced in all patients.

Stereopsis was absent in all patients.

### 3.3. Ophthalmological Assessment

No abnormalities in the anterior segment were found, and no corneal nor lens opacity was observed. Slight optic nerve head pallor was observed in all but one patient. No other retinal abnormalities were recorded. Second-level ophthalmological examination (OCT; retinography) could not be carried out because of lack of cooperation. Hyperopia was found in two patients and myopia in two, associated with mild astigmatism. In three patients, fundus evaluation was not performed by means of pupil mydriasis due to lack of cooperation or unstable general conditions. Mydriatic eye drops could only be administered to four patients. In one patient, a previous side effect (dyskinetic status) after eye-drop administration was recorded. All patients showed moderate photophobia.

### 3.4. Anamnestic Data Collection

#### 3.4.1. Neuroimaging Features

Six out of seven patients underwent brain magnetic resonance imaging (MRI) at 1.5 Tesla. MRI was normal in five of the six patients and evidenced periventricular hyperintensity in the remaining one.

#### 3.4.2. Genetic Data

All children showed pathogenic GNAO1 variants (Table 1), confirmed with a custom enrichment panel for more than 100 genes related to pediatric MDs or a panel of next-generation sequencing for epilepsy, performed in several neurogenetics laboratories. Four out of seven mutations were gain; two were unknown functions; and the remaining one was normal function.

## 4. Discussion

GNAO1-related encephalopathy is a rare condition with a broad phenotypic spectrum characterized by cognitive delay [7], motor and communicative difficulties, epileptic disorder and movement disorders (MDs).

The possibility to perform a complete visual assessment is generally limited in patients with genetically determined encephalopathy due to poor cooperation, movement disorders and reduced verbal ability, which make it difficult to define the response. Most information on visual competences comes from the ophthalmologic assessment or from caregivers’ observation.

This study reports, for the first time, a detailed assessment of visual function in children with GNAO1 mutations, which was carried out using a battery of tests specifically designed for young patients with relatively poor cooperation that are easy to perform even in subjects with multisensory/cognitive impairment [15,18].

Our data show that visual function was impaired in all patients. Less complex aspects of visual function appeared to be normal in all but one patient. These children were able to fix and track horizontally and vertically but had some difficulties in more complex actions, such as tracking in a circle. It is noteworthy that the organization of saccades was preserved horizontally in all children but one, while vertical exploration was impaired in more than 50% of subjects. Oculomotor abilities appear to be more related to object–face exploration, recognition and environmental control mediated through the ventral pathway, creating a complex archive of visual data in the temporal lobe. Recalling this information is crucial to process visual inputs and may influence visuo-cognitive function and social life. Our data are in accordance with previous studies on the maturation of global visual motion sensitivity. This sensitivity has been described as delayed when compared with global visual form sensitivity across a diverse range of neurodevelopmental disorders, such as developmental dyslexia, autism spectrum disorders, Williams syndrome, Fragile X syndrome, prematurity, hemiplegia and congenital cataract, suggesting “dorsal stream vulnerability” [27,28,29,30].

More cortical aspects of visual function, such as visual acuity, visual field, contrast sensitivity and stereopsis, were also impaired in almost all patients. More specifically, all patients presented with poor ocular motility, low vision, visual acuity <3/10 and absence of stereopsis. Visual fields and contrast sensitivity were reduced in six of the seven patients. These aspects of visual function were impaired in all, irrespectively of the age at assessment. These functions collect information about discrimination at distance, perception of distance, depth and volume.

These competences are important for movement planning and for the understanding of space. The impairment of these aspects of visual function could influence, in association with hypotonia and movement disorder, the acquisition of postural and motor milestones.

It is of interest that even if acuity, fields and contrast sensitivity were often impaired, the residual vision was enough to achieve good near vision. This implies the possibility to develop a good ability to discriminate and process visual information at a close distance.

The visual difficulties that these patients presented with appeared to be quite similar irrespectively of epilepsy (onset and severity), type of movement disorders and type of genetic mutation.

Only one patient (Pt 6) had more global visual impairment, involving ability to fix and track, and obtained the worst results of all in visual acuity, attention at distance and contrast sensitivity.

An early assessment of visual function can, therefore, be of help to define the abilities of patients with GNAO1 mutations in order to more accurately choose specific devices for communication and learning.

Since the description of visual competences was the result of a single observation and since visual competences might be expressed in different ways in more familiar contexts and everyday life, a longitudinal observation in a larger cohort would be advisable to confirm the results.

## 5. Conclusions

Our data suggest that in children with GNAO1 mutations, several visual aspects are impaired. The early recognition of these difficulties might lead to early rehabilitation specifically targeting visual function.

Visual impairment did not appear to be grossly related to the presence of other clinical signs, such as epilepsy or MDs. Further studies using a more global assessment could help to establish whether this mild but specific impairment of distinct aspects of visual function may have an effect on neurodevelopmental and cognitive abilities.

Performing the assessment on subjects with a milder phenotype might be of help to define a correlation between the severity of clinical presentation and visual impairment.

Early assessment of visual abilities can give information on the possibility to use these competences for communication and learning strategies and to define a more focused rehabilitation program. Furthermore, longitudinal assessment could help to understand whether the maturation of some visual aspects may be foreseeable after specific training.

## Figures and Tables

**Table 2 genes-14-00544-t002:** Visual function assessment.

	Pt 1	Pt 2	Pt 3	Pt 4	Pt 5	Pt 6	Pt 7
Age/Sex	2 y/F	3 y/F	3 y/M	5 y/F	6 y/M	7 y/M	8 y/M
Eye movement	Ocular dyspraxia	Ocular dyspraxia	Ocular dyspraxia	Ocular dyspraxia	Ocular dyspraxia	Strab, nys	Ocular dyspraxia
Fixation	Stable	Stable	Stable	Stable	Stable	Unstable	Stable
Horizontal tracking	Complete	Complete	Complete	Complete	Complete	Incomplete	Complete
Vertical tracking	Complete	Complete	Incomplete	Complete	Incomplete	Incomplete	Incomplete
Attention at distance	300 cm	300 cm	300 cm	300 cm	300 cm	70 cm	300 cm
Horizontal saccades	Present	Present	Present	Present	Present	Absent	Present
Vertical saccades	Present	Absent	Present	Present	Absent	Absent	Absent
Visual acuity	1.2/10	2/10	1/10	2.5/10	1.2/10	1/50	1.2/10
Visual field	Bilateral reduction	Bilateral reduction	Normal	Bilateral reduction	Bilateral reduction	No response	Bilateral reduction
Contrast sensitivity	Reduced	Reduced	Reduced	Reduced	Reduced	Reduced	Reduced
Stereopsis	Absent	Absent	Absent	Absent	Absent	Absent	Absent

Y, years; F, female; M, male; Strab, strabismus; Nys, nystagmus.

## Data Availability

The data that support the findings of this study are available upon request from the corresponding author.

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
