# Peer review of "Visual Function in Children with GNAO1-Related Encephalopathy"

_genes, 2023, doi:10.3390/genes14030544_

Round 1

Reviewer 1 Report

This is a relatively small study on GNAO1 patients all severely disabled in this group trying to assess their visual function .

The assessment is quite comprehensive and evaluate a wide range of basic visual functions in a mothod the author claim to be compatible to perform in this kind of population.

although a lot of effort was mode to get responses which can be assessed there is still an option that the level of alertness of the patients (especially the ones on ASM's ,Their general attention level, their primary movement disorder makes part of the tests chosen to be unreliable while in real life the function assessed is either better or exist while not picked up during the test, This limitation should be mentioned in the discussion section .

I am also concerned about the conclusion and generalization made related to the visual pathway affected in these patients based on the possible fluctuation in their response and the relatively small number of patients assessed .In order to confirm this assumption and generalize it to GNAO1 patients children with milder presentation should be also assessed or at least it should be mentioned in the discussion  

Author Response

This is a relatively small study on GNAO1 patients all severely disabled in this group trying to assess their visual function .

The assessment is quite comprehensive and evaluate a wide range of basic visual functions in a mothod the author claim to be compatible to perform in this kind of population.

although a lot of effort was mode to get responses which can be assessed there is still an option that the level of alertness of the patients (especially the ones on ASM's ,Their general attention level, their primary movement disorder makes part of the tests chosen to be unreliable while in real life the function assessed is either better or exist while not picked up during the test, This limitation should be mentioned in the discussion section.

Response 1: We agree that in a single observation there is the possibility to fail to pick up the best performance of each patient. We added a sentence in discussion.

I am also concerned about the conclusion and generalization made related to the visual pathway affected in these patients based on the possible fluctuation in their response and the relatively small number of patients assessed .In order to confirm this assumption and generalize it to GNAO1 patients children with milder presentation should be also assessed or at least it should be mentioned in the discussion  

Response 2: We agree that having a larger cohort might help to describe different severity of visual impairment according to different phenotype. We added a sentence in conclusions.

Reviewer 2 Report

This manuscript presents results of original study in the field of phenotypic expression of monogenic disorders. The study was conducted with appropriate methodology, results are clearly presented and critically discussed. The topic was chosen with the aim to better characterize the phenotype of the GNAO1 related encephalopathies spectrum, especially to describe the profile of neurovisual competences in affected children. This group of encephalopathies include a broad spectrum of developmental disorders caused by de novo heterozygous mutations in the GNAO1 gene, encoding the G (o) subunit α of G-proteins. In presented study authors have analyzed group of seven patients with confirmed genetic diagnosis i.e. GNAO1 mutation, who underwent a precisely described neurological and ophthalmological evaluation.  The results showed that in children with GNAO1 related encephalopathies visual function is often impaired with pattern of visual impairment that are less affecting the ventral stream. The significance of this research is that it points out to the need of early recognition of these difficulties which is important as it may lead to early rehabilitation specifically targeting visual function. Visual impairment did not appear to be grossly related to the presence of others clinical signs such as epilepsy or movement disorders. In my opinion this article has importance for clinical practice and for patients. 

Author Response

Response 1: Thank you for your attention and recognition of the value of this work which aims to be a small contribution to a better definition of the phenotype of GNAO1-related diseases.